# Arsenic Trioxide and (−)-Gossypol Synergistically Target Glioma Stem-Like Cells via Inhibition of Hedgehog and Notch Signaling

**DOI:** 10.3390/cancers11030350

**Published:** 2019-03-12

**Authors:** Benedikt Linder, Andrej Wehle, Stephanie Hehlgans, Florian Bonn, Ivan Dikic, Franz Rödel, Volker Seifert, Donat Kögel

**Affiliations:** 1Experimental Neurosurgery, Department of Neurosurgery, Neuroscience Center, Goethe University Hospital, 60528 Frankfurt am Main, Germany; Andrej.Wehle93@gmx.net (A.W.); koegel@em.uni-frankfurt.de (D.K.); 2Radiotherapy and Oncology, Goethe University Hospital, 60590 Frankfurt am Main, Germany; Stephanie.Hehlgans@kgu.de (S.H.); Franz.Roedel@kgu.de (F.R.); 3Institute of Biochemistry II, Faculty of Medicine, Goethe University Hospital, 60590 Frankfurt am Main, Germany; bonn@med.uni-frankfurt.de (F.B.); dikic@biochem2.uni-frankfurt.de (I.D.); 4Buchmann Institute for Molecular Life Sciences, Goethe University, 60438 Frankfurt am Main, Germany; 5Department of Neurosurgery, Goethe University Hospital, 60528 Frankfurt am Main, Germany; V.seifert@em.uni-frankfurt.de; 6German Cancer Consortium (DKTK), Partner Site Frankfurt, 60590 Frankfurt am Main, Germany

**Keywords:** cancer stem cells, glioblastoma, Hedgehog, Notch, DNA damage

## Abstract

Glioblastoma is one of the deadliest malignancies and is virtually incurable. Accumulating evidence indicates that a small population of cells with a stem-like phenotype is the major culprit of tumor recurrence. Enhanced DNA repair capacity and expression of stemness marker genes are the main characteristics of these cells. Elimination of this population might delay or prevent tumor recurrence following radiochemotherapy. The aim of this study was to analyze whether interference with the Hedgehog signaling (Hh) pathway or combined Hh/Notch blockade using small-molecule inhibitors can efficiently target these cancer stem cells and sensitize them to therapy. Using tumor sphere lines and primary patient-derived glioma cultures we demonstrate that the Hh pathway inhibitor GANT61 (GANT) and the arsenic trioxide (ATO)-mediated Hh/Notch inhibition are capable to synergistically induce cell death in combination with the natural anticancer agent (−)-Gossypol (Gos). Only ATO in combination with Gos also strongly decreased stemness marker expression and prevented sphere formation and recovery. These synergistic effects were associated with distinct proteomic changes indicating diminished DNA repair and markedly reduced stemness. Finally, using an organotypic brain slice transplantation model, we show that combined ATO/Gos treatment elicits strong growth inhibition or even complete elimination of tumors. Collectively, our data show for the first time that ATO and Gos, two drugs that can be used in the clinic, represent a promising targeted therapy approach for the synergistic elimination of glioma stem-like cells.

## 1. Introduction

Glioblastoma (GBM, WHO grade IV astrocytoma) is the most common and most aggressive primary brain tumor in adults [1]. Due to its diffuse infiltrative growth, surgical resection is virtually impossible [2]. Thus, even with the current standard-of-care treatment consisting of temozolomide (TMZ)-based radiochemotherapy [3], the average survival barely exceeds one year and the 5-year-survival rate is below 5%. Another key characteristic of GBM is its high resistance to induction of apoptosis, which is in part mediated by overexpression of anti-apoptotic proteins and an enhanced DNA repair (reviewed in [4]).

Many cancers including GBM exhibit a hierarchical organization of cells including a population with a stem-like phenotype (GSCs: glioma stem cells) that can replenish the tumor after treatment and are thought to be responsible for disease recurrence (reviewed in [5]) and initiation [6]. GSCs harbor expression of stemness-related marker genes and phenotypical traits similar to normal stem cells (reviewed in [5]) including an unlimited regenerative potential and the ability to divide asymmetrically. It was further shown that GSCs are highly resistant to DNA damaging agents and ionizing radiation and have an increased DNA repair capacity, in part caused by enhanced expression of Checkpoint kinase 1 (CHK1) and CHK2 [7]. 

The Hh and Notch signaling pathways are developmentally important signaling pathways involved in body patterning, brain development and maintenance of stem cells [8]. During the last decade it became apparent that Hh signaling plays a crucial role for tumor formation and progression of many cancers including GBM ([9] and reviewed in [10]). Notch signaling is also involved in brain development and regulates the expression of a wide variety of target genes that increase cell proliferation and survival (reviewed in [11]). Recent research also revealed a central role in self-renewal and therapy resistance of GSCs ([12] and reviewed in [13]), indicating that Notch represents a potential therapeutic target in these cells. 

Arsenic trioxide (ATO) was initially used in Traditional Chinese Medicine to treat various diseases, including cancer. Research based on these findings resulted in the application of ATO as a clinical drug for treatment of acute promyelocytic leukemia (APL). Further, many studies have demonstrated potent Notch-inhibitory and antitumoral effects of ATO in various other cancer entities [14] including glioma [15]. In addition, ATO has been shown to effectively inhibit Hh signaling at the level of GLI in multiple cancers including medulloblastoma [16].

Based on the findings that both pathways regulate key aspects of GSCs, we aimed at investigating the potential therapeutic effects of Hh and Notch inhibition in regard to stem cell differentiation and sensitivity to cell death induced by (−)-gossypol (AT-101, Gos), a natural compound that efficiently induces cell death in apoptosis-resistant, differentiated GBM cells [17]. Recent research indicates that Gos also targets cancer and cancer stem cells (CSCs) in colorectal [18] and prostate cancer [19], respectively. We hypothesized that an enforced differentiation of GSCs might render them more vulnerable to cell death induction, and that a combination of Hh and/or Notch inhibition consequently would synergistically increase the effects of Gos. We further aimed to scrutinize whether single pathway inhibition is sufficient in driving cell death or whether multiple intertwined (i.e., Hh and Notch) pathways need to be targeted. To this end, we employed the well-known Hh pathway inhibitor/GLI-antagonist 61 (GANT61, GANT; [20]) and ATO in combination with Gos.

Here, we demonstrate that ATO-mediated Hh/Notch inhibition together with the natural anticancer agent Gos, rather than combining the Hh-inhibitor GANT and Gos efficiently blocks GSC growth, induces cell death and diminishes the self-renewal capacity of tumor sphere lines. These therapeutic effects are accompanied by distinct proteomic changes suggesting alterations in cell movement and cell cycle progression. They are also associated with major alterations in the expression of (neuronal) development genes (e.g., oligodendrocyte transcription factor 2 (OLIG2), sex determining region Y (SRY)- box 2 (SOX2) and SOX9) and DNA damage repair genes (e.g., BRCA1 Associated ATM Activator 1 (*BRAT1*), CHK1, CHK2), suggesting that this drug combination can efficiently enforce GSC differentiation and sensitization to therapy. Finally, this treatment was applied to tumors transplanted onto adult organotypic brain slices where ATO and ATO/Gos decreases tumor size and even lead to the complete elimination of some tumors.

## 2. Results

### 2.1. Hh/Notch Inhibition in Combination with Gos Synergistically Induces Cell Death of GSCs

The general aim of this study was to identify novel approaches to selectively target GSCs using established small molecule inhibitors to inhibit either the Hh signaling pathway alone or in combination with the Notch pathway. To this end, we applied GANT or ATO to target the Hh or Hh/Notch signaling pathways, respectively, in combination with the cell death inducer Gos. Our hypothesis was that the enforced differentiation mediated by either Hh or Hh/Notch inhibition would increase the vulnerability of GSCs towards cell death stimulation with Gos, resulting in synergistic action of the two drugs. To test this hypothesis, we initially performed MTT assays (Figure 1) and calculated the combination indices (CIs) according to Chou et al. [21] (CI < 1: synergism; CI = 1: additive; CI > 1: antagonism). Morphologically, a reduction of sphere size and apparent disintegration of the spheres after single agent treatment with GANT, ATO or Gos was detectable, indicated by an increase of single cells (black arrows) and less tight appearance of the spheres (white arrows). Combination treatment further enhanced these effects (Figure 1a).

MTT assays with the tumor sphere line GS-5 (Figure 1b,c) showed that single agent treatment with GANT, ATO or Gos dose-dependently reduced the viability and combination treatments synergistically enhanced these effects (CI < 1). Similar findings were also made with the GANT/Gos and ATO/Gos combinations in GS-1 cells (Appendix A), and with the GANT/Gos, but not ATO/Gos combination in GS-8 cells (Appendix A), although GANT single agent treatment had no significant effects in these cells. 

The decreases in viability were affirmed by increases in cell death as shown by FACS-based Annexin V/Propidium iodide (PI) double stainings (Figure 1d–f). Again the combination treatments were more effective than either single treatment. Similar findings were also made in two other GS-lines (GS-3 and GS-8, Appendix A) and a GS-line with a restricted stem-like (progenitor-like) phenotype (GS-1, Appendix A).

Next, we analyzed the expression of *GLI1*, *PTCH1*, *BCL2* and *CCND1*(Cyclin D1) as surrogate markers for Hh pathway activity as well as *HES5* and *HEY1* for Notch signaling in GS-5 (Figure 1g) and the primary culture 17/02 (Figure 1h). Despite the fact that we applied GANT at 2.5 µM, a concentration that exhibits robust inhibitory activity of Hh signaling in the Gli-responsive cell line Shh light II [22] (Appendix A), it had little effect on any of the analyzed target genes, although a small tendency towards *PTCH1* and *CCND1* inhibition was apparent. Gos alone strongly reduced *PTCH1* and *HES5* expression. *HES5* expression was also reduced after GANT + Gos treatment. ATO and ATO + Gos reduced the expression of all markers, except *HEY1* in 17/02, whereas the combination exerted greater inhibitory effects. Similar findings were also observed for GS-8 and a second primary culture, 17/01. Notably, 17/01 appeared to be insensitive towards Hh-inhibition and only showed minor inhibition of the Notch-targets. Curiously, we observed that Gos increased the expression of *GLI1* in GS-5, GS-8 and 17/02, while simultaneously decreasing *PTCH1*-expression.

### 2.2. Proteomic Analysis Reveals Global Changes Related to Impaired Cell Movement, DNA Repair and Stemness Properties after ATO/Gos Treatment

To analyze the underlying processes mediating the observed synergistic antitumoral effects in more detail, we next performed unbiased proteomic analyses using label-free quantification of GS-5 cells (Figure 2). For our proteomics approach we chose lower drug concentrations that did not result in massive cell death (2.5 µM GANT/ATO each and 5 µM Gos). Using this approach, we obtained reproducible quantitative data for 5008 proteins. Compared to solvent-treated cells 255, 47 and one protein(s) differed significantly after single treatment with Gos (Figure 2a), ATO and GANT (Appendix A), respectively. The combination treatments markedly elevated the amount of significant changes found to 424 and 648 after GANT/Gos (Figure 2c) and ATO/Gos (Figure 2e) treatment, respectively, thereby further providing evidence for the observed drug synergism. The five most increased and decreased proteins for each condition are depicted in the Volcano Plots (Figure 2a,c,e) along with the known stemness markers OLIG2, SOX2 and SOX9. The decreased proteins include proteins with known functions for glioma stemness and migration/invasion like CD9 [23] (Gos, ATO/Gos), Ephrin receptor A2 (EPHA2 [24]; Gos, GANT/Gos) and mesenchymal (i.e., more aggressive) differentiation like FRAS1 related extracellular matrix protein 2 (FREM2 [25]; ATO/Gos). We validated selected genes among the top five hits via qRT-PCR (Appendix A). A list displaying all significantly changed proteins is provided in the Appendix A.

To analyze for Gene Ontology (GO) terms (Figure 2b,d,f) that are enriched among the changed proteins we used the STRING-platform, (string-db.org; ver. 10.5; accessed Feb. 2018; [26]) with a consecutive analysis using Revigo (revigo.irb.hr; accessed Sep. 2018; [27]). Treatment with Gos (Figure 2b) caused a decrease in proteins related to (neuronal) development and movement, while proteins from the translation machinery were overrepresented among the increased proteins. The combined treatment of GANT and Gos (Figure 2d) enriched for cell cycle- and movement-related proteins among the decreased proteins. Among the increased hits, proteins involved in the processes “response to oxidative stress”, “cellular respiration” and “response to unfolded protein” are overrepresented.

Combined ATO and Gos treatment (Figure 2f) caused an even stronger enrichment of (neuronal) development proteins among the significantly decreased proteins. This group includes the known GSC markers OLIG2, SOX2 and SOX9. In addition, movement and cell cycle proteins are found to a large portion in decreased amounts. Similar to the single treatment with Gos the increased proteins were often assigned to translation and again “response to oxidative stress” and “cellular respiration”. A list displaying all enriched GO:BP terms is provided as a Appendix A.

Next we analyzed the degree of overlap between the three groups (Figure 2g, Appendix A). Here, 53 and 45 proteins are significantly decreased and increased following all three treatments, respectively. 46 and 146 proteins are decreased and increased only by the combination treatments (GANT/Gos, ATO/Gos), but not by Gos alone. Notably, proteins decreased only after both combination treatments include proteins involved in the DNA damage response (DDR) like BRAT1, Fanconi Anemia Complementation Group I (FANCI) and CHEK1, indicating that the cells might be more prone to DNA damage. GO-analysis (Appendix A) of the proteins regulated by both combination treatments revealed that only the GO:BP term mitotic cell cycle is significantly enriched among the decreased proteins. Some terms were assigned to translation, cellular respiration, and response to oxidative stress among the increased proteins. 

### 2.3. ATO and Gos Treatment Reduces Expression of Stemness Markers 

A key hallmark of GSCs is the expression of stemness-related genes like OLIG2, SOX2 or SOX9 with a concomitant lack of known differentiation genes like GFAP. Indeed, our proteomic analysis showed that OLIG2 abundance decreases upon GANT/Gos and ATO/Gos treatment, while ATO/Gos also significantly decreased SOX2 and SOX9 protein levels and increased GFAP.

Western Blot (Figure 3a,b) analyses confirmed that all treatments except GANT alone reduce OLIG2 and SOX9 expression in GS-5 cells (Figure 3a) with ATO and the ATO/Gos combination being most effective. Additionally, SOX2 was only decreased after ATO treatment. ATO also strongly reduced OLIG2, SOX2 and SOX9 mRNA expression which was further enhanced by Gos (Figure 3c). In 17/02 especially ATO and ATO/Gos reduce the protein levels (Figure 3b) of the three markers, while all treatments except GANT reduced the expression of the corresponding mRNAs (Figure 3d). In GS-8 (Appendix A) we obtained similar data compared to 17/02, although the mRNA expression was only changed after ATO or ATO/Gos treatment and 17/01 (Appendix A) was mainly reacting to ATO/Gos treatment. In order to analyze expression changes on a single cell level we performed immunofluorescence microscopy of GS-5 and could show that all treatments except GANT can reduce the amount of marker positive cells with the combination treatment being the most effective.

### 2.4. ATO and Gos Treatment Induces DNA Damage Via Downregulation of DDR Genes

A key hallmark of GSC is their treatment resistance towards conventional chemotherapy by enhanced DNA repair, which is in part facilitated by overexpression of CHK1 and CHK2 [7]. Interestingly, CHK1 was significantly decreased according to our proteomic data. This finding prompted us to analyze additional key targets involved in the DNA damage response (DDR) including *CHK1*, *CHK2*, *RAD51* and Survivin (*BIRC5*) (Figure 4a). Survivin is a known downstream target of both Hh [28] and Notch signaling [29] and has been linked to DNA damage [30] and radiation resistance of GBM [31]. Accordingly, we observed that GANT/Gos and ATO/Gos significantly decreased *CHEK1*, *CHEK2*, RAD51 Recombinase (*RAD51*) and *BIRC5* (Survivin) expression, while ATO/Gos also decreased *BRAT1* and Ataxia Telangiectasia Mutated (*ATM*) expression. Similary, CHK1, CHK2 and Survivin protein levels were also decreased upon treatment, most pronouncedly after ATO and ATO/Gos treatment. This was also accompanied by loss of CHK1 phosphorylation (Figure 4b), while phosphorylated CHK2 could not be detected. Since functional loss of CHK1 and CHK2, two main effectors of the DDR might lead to accumulating DNA damage, finally resulting in cell death, we next analyzed DNA damage induction after drug treatment by immunofluorescent microscopy of TP53BP1- and γH2AX-positive foci (Figure 4c–e).

All single treatments significantly increased the number of TP53BP1- (Figure 4c) and γH2AFX-positive foci (Figure 4d) in GS-5, which could even be increased using the combination treatment. Of note, the increase in γH2AFX foci did not reach statistical significance for Gos and GANT alone. 

Strikingly, the amount of TP53BP1-positive foci of the combination treatment is significantly higher than either single treatment, indicative of synergism. As a visual control for DNA damage/foci induction the cells were also treated with Etoposide (Figure 4e), a known inducer of DNA damage. Similar findings were also observed in GS-3 (Appendix A), while GS-8 only showed detectable induction of DNA damage after ATO and ATO/Gos treatment. 

### 2.5. Effects of ATO and Gos on Sphere Forming Capacity and Stem-Cell Frequency of GSCs

Another key hallmark of GSCs is the ability to form new spheres from single cells in vitro [32]. Furthermore, our proteomic analyses clearly showed that multiple GO-terms related to neuronal differentiation and development are enriched among the decreased proteins following ATO/Gos treatment. In order to functionally test whether the treatment indeed reduces stemness properties, we performed limiting dilution assays (LDA; Figure 5a,b) and extended sphere formation assays (SFA [33]; Figure 5c–f) (for a schematic overview see Figure 5c).

The LDA revealed that treatment with ATO and ATO/Gos treatment reduced the stem-cell fraction about 15- and 7-fold in GS-5 and 17/02, respectively, whereas GANT or GANT/Gos treatment only reduced the stem-cell fraction 3-fold in GS-5 and only slightly in 17/02. In contrast, Gos did not change the ratio of sphere-forming vs total cells. GS-8 reacted similar to GS-5 (Appendix A).

Next, we performed the extended SFA (Figure 5d–f). We determined primary sphere formation (Figure 5d), sphere recovery (Figure 5e) and secondary sphere formation (Figure 5f) of GS-5 cells after treatment with lower doses of the drugs (1 µM ATO, 2.5 µM GANT and 0.5 µM Gos). Treatment with Gos did not change primary sphere formation or sphere recovery. GANT or GANT/Gos led to smaller spheres, but did not reduce sphere number. In fact, we observed more primary spheres after GANT and GANT/Gos treatment. In comparison, ATO treatment was far more effective as it resulted in significantly smaller and fewer spheres, even in the secondary sphere formation assay. Similar findings (no to low effects of Gos, moderate effects of GANT, strong effects of ATO) were also observed in GS-3 (Appendix A) and GS-8 (Appendix A). Here we could show that ATO/Gos strongly decreases the stem-cell frequency and disturbs sphere formation and recovery.

### 2.6. ATO and ATO/Gos Treatment Inhibits Tumor Growth in Adult Organotypic Slice Cultures

To finally test the proposed therapeutic approach in a more complex and pathophysiological relevant model, we used adult organotypic brain slices (OTC), transplanted GFP-Luc-positive GS-5 (Figure 6), GS-8 (Appendix A) or CellTracker CM Dil-labeled 17/02 (Appendix A) cells onto these OTCs and analyzed tumor growth under treatment over time. We observed that GANT or Gos single agent treatment of GS-5 tumors partially affected the growth of the tumors, but did not cause any reduction of tumor sizes over time. The GANT/Gos treatment resulted in a fast (within 6 days) decrease of tumor sizes, and these effects remained stable until the end of the experiment (28 days). In contrast, treatment with ATO, and even more profoundly with ATO/Gos, continuously decreased the tumor sizes until only small tumor cell populations remained, while other tumors disappeared entirely (Figure 6a, white arrows). 

The GS-8 tumors were more sensitive towards ATO and ATO/Gos, while only showing a growth retardation using GANT and GANT/Gos. Finally, we observed that the patient-derived primary culture 17/02 shows growth retardation after treatment with ATO, Gos and GANT/Gos, and tumor sizes remained almost at baseline after ATO/Gos treatment. It should be emphasized that the primary culture 17/02 was derived from a heavily pre-treated patient (see materials and methods), likely associated with an aggressive phenotype and high therapy resistance of the corresponding tumor. Taken together we can show that ATO and the ATO/Gos drug combination can effectively reduce tumor growth or even eliminate tumors in the absence of observable detrimental effects in the surrounding brain tissue in an organotypic environment.

## 3. Discussion

Cancer recurrence usually results in more aggressive and treatment-resistant disease and finally in the death of the patient. Since diffusely infiltrating GBMs cannot be completely resected and a fraction of residual cancer cells are thought to exhibit or obtain a stem-like [35] phenotype, disease recurrence is virtually unavoidable. Previous research showed that when GSCs are enforced to differentiate through cultivation in fetal calf serum (FCS) containing media, they are more susceptible to conventional chemotherapy [36]. Hence, a proposed strategy for eliminating GSCs is to enforce their differentiation. Thus, we inquired whether small molecule-mediated inhibition of the Hh signaling pathway, which has been described as a major driver of GSCs in several studies [9,37], is sufficient for depleting this cell population in combination with the natural anticancer agent AT-101/Gossypol (Gos) [17], or if additional inhibition of other pathways implicated in therapy resistance of GBM, in particular Notch-signaling [13,33], does so more efficiently. 

Here, we show for the first time that although the Hh inhibitor GANT and cell-death inducer Gos synergistically induce cell death, only combined Hh/Notch inhibition using ATO in combination with Gos is able to efficiently target multiple GSC lines and two primary GBM cultures in vitro and ex vivo. Other groups have described Hh inhibition alone to be sufficient for targeting GSCs [9,37]. However, we did not observe a robust depletion of GSC-properties by the Hh inhibitor GANT in our lines. Nonetheless, we regularly observed minor effects of single agent treatment with GANT, including a reduction of cell viability, induction of cell death and DNA damage, as well as depletion of sphere forming cells, as described by other groups using genetic models of Hh inhibition [9,37,38]. How could these rather moderate effects of GANT in our experiments be explained? Notably, one major aim of this study was to assess the synergy of combination treatments, which prompted us to use lower concentrations of all drugs when used as single agent treatments. Although Hh/GLI-signaling has been described to be essential for GBM [39], so far only one study reported the use of GANT [40] using FCS-cultured GBM cells. In the study by Li et al. [40] relatively high concentrations (up to 10 µM) and treatment periods (up to 72 h) were applied. Considering that GANT61 requires a hydrolysis step to yield the biologically relevant form of the drug [41], it is also possible that this step is disturbed or can even be reversed in glioma stem-like cells. This proposition should be analyzed in future studies. However, considering that ATO alone already depleted the GSC population in this and other studies [12] quite effectively, while reducing the expression of known Hh and Notch target genes, we conclude that the ATO/Gos combination is superior to GANT/Gos and that ATO is the more potent, although less specific, Hh-pathway inhibitor. In any case, our data supports our main hypothesis, i.e., that multiple intertwined signaling events need to be targeted to effectively eliminate GSCs.

Our proteomic data suggested the disturbance of processes related to cell movement, highlighted by decreased levels of several proteins involved in glioma migration, such as CD9 [23] and EPHB3 [42] after treatment with Gos or ATO/Gos, as well as EPHA2 after treatment with GANT/Gos. More prominently, we also found processes related to the cell cycle/DDR and to neuronal development to be disturbed. These changes indicate a reduced capability of tumor invasiveness/migration. Additionally, the proteomic analysis revealed an increase of many proteins involved in cellular respiration (GO.0045333) and the response to oxidative stress (GO.006979) after GANT/Gos and ATO/Gos treatment. In line with these observations, Gos [43], ATO [44] and GANT61 [45] have been described to impair mitochondrial function and to induce ROS-formation in various settings. 

Based on our proteomic data, we next analyzed whether the expression of key mRNAs/proteins of the DNA damage response may be altered following drug treatment. Indeed, we could show a pronounced reduction of CHK1/*CHEK1*, CHK2/*CHEK2* and *RAD51* expression after both GANT/Gos and ATO/Gos, and additionally of *ATM*, *BRAT1* and *FANCI* after ATO/Gos combination treatment. The facts that (1) an enhanced DNA repair capacity is an established hallmark of GSCs [7] and (2) we observed increased DNA damage in three GSC lines with the concurrent reduction in DDR proteins/genes lend further support to the notion that the ATO/Gos combination efficiently depletes stem-like cells by reversing their phenotype and increasing their susceptibility for cell death stimulation. Notably, a central role of Hh (reviewed in [46]) and Notch signaling [47] in regulating the DDR has been proposed. Accordingly, inhibition of Notch1 increased the radiosensitivity of TALL-1 T-cell acute lymphoblastic leukaemia cells in vitro [47] and inhibition of Hh signaling in U87 glioma cells [48] increased their radiosensitivity. Interestingly, Gos has been shown to induce ROS-mediated DNA damage in cancers such as gastric cancer [49], where it also disturbed self-renewal of CD133-positive cells. Furthermore, Gos induced DNA damage and cell death in prostate cancer cell lines, stem-like cells and in vivo xenografts [19]. Our findings further imply that the ATO/Gos treatment sensitizes GSCs to chemo- and/or radiation therapy. Thus, future research should also be directed towards analyzing the combination of ATO/Gos with ionizing radiation, preferably in an in vivo setting.

Most importantly, we detected reduced expression of the stemness marker OLIG2, SOX2 or SOX9, after ATO/Gos treatment in two GS lines and two primary patient-derived cultures. The GANT/Gos combination did not result in such pronounced effects compared to ATO/Gos. These data suggest that dual inhibition of Hh and Notch signaling is superior to Hh inhibition alone, indicating that the stem-like phenotype is maintained via input from multiple signaling pathways. Our data also support the notion that ATO enforces differentiation of GSCs, while Gos seems to be dispensable for limiting stemness as it did not reduce stem-cell frequency or sphere formation in the phenotypic assays. Counterintuitively, our proteomic data suggested that Gos reduced developmental processes, whereas ATO did not. This apparent discrepancy may in part be explained by the lower concentrations of ATO used for the proteomic screen (i.e., 2.5 µM) in comparison to the western blot analyses (i.e., 5 µM), although the lower concentration used for the proteomic screen was also applied for the qRT-PCR analyses, where ATO clearly reduced stem marker expression. At the applied concentration (5 µM), Gos reduced proteins related to stemness in the proteomic dataset which could be confirmed in our qRT-PCR and Western Blot analyses. In line with these observations Gos is capable to reduce stem-cell markers in stem-like prostate cancer cells [19]. However, these proteomic changes appear to be insufficient to evoke a phenotypic response, since we observed neither changes in stem-cell frequency nor in sphere formation could be observed. From these data we conclude that ATO is the drug primarily serving to de-differentiate GSCs following combined ATO/Gos treatment, while Gos enhances these effects. 

Aside from the effects on differentiation, ATO is also known to regulate a variety of unrelated processes that likely contribute to the depletion of GSCs. Accordingly, Sun et al. showed that ATO induces ROS formation in rat glioma cells thereby inducing apoptosis [50]. In line with these findings, we observed increased DNA damage after ATO or Gos treatment, and even more pronounced after the combination treatment, likely caused by excessive ROS formation [51]. This is also reflected in our proteomic analysis showing increased expression of proteins of the GO-BP-term “response to oxidative stress” (GO.0006979). Conclusively, the most increased protein after ATO and ATO+Gos treatment is heme oxygenase 1 (HMOX-1). Of note, HMOX-1 is believed to mediate ATO-induced mitochondrial damage in rat astrocytes [52]. In fact, we could recently show that HMOX-1 is induced after Gos-mediated mitochondrial dysfunction in non-GSC glioma cells and that this is accompanied by induction of BNIP3 and Nix (BNIP3L) [53], leading to an autophagic type of cell death. Strikingly, a similar mechanism (BNIP3/BNIP3L-mediated autophagic cell death) has also been proposed for ATO in non-GSC glioma cells [54], indicating that Gos and ATO might target overlapping pathways to promote cell demise of GSCs.

Following this line of thougt, we also observed an increase of proteins of the GO-BP term “response to unfolded protein” (GO.0006986). Interestingly, these alterations may in part reflect the binding properties of ATO to target proteins via thiol-adduction of cysteine-residues. For example, this has been shown in acute promyelocytic leukemia (APL), where ATO is already used in the clinic, and binds to PML and the PML-RARα fusion protein that is found in almost all APL-cells [55]. This binding is thought to disturb the correct protein folding and function, thereby inducing target degradation via induction of the unfolded-protein response [56,57]. ATO was additionally shown to bind to thioredoxins and glutathione, proteins that are involved in cellular respiration and the antioxidant defense, respectively (reviewed in [58]), which may further contribute to ROS formation and cell death. 

Notch signaling was shown to enhance Hh signaling in neural stem cells by modulating the ciliary trafficking of the canonical Hh pathway component Smoothened (Smo) by regulating the presence of Patched [59]. Thus, the blockade of Notch through ATO might target Hh signaling by multiple mechanisms: either directly at the level of Gli and/or more upstream by preventing ciliary entry of Smo. Accordingly, we detected an enrichment of the GO-term “regulation of cell projection organization” (GO.0034344) after treatment with ATO/Gos, indicative of reduced ciliary formation. In analogy to our findings, it was shown that combined Hh and Notch inhibition using the archetypical Smo inhibitor cyclopamine in combination with the gamma-secretase inhibitor GSI-1 induced cell death in CD133-enriched glioma cells and increased the cytotoxicity of TMZ [60]. Strikingly, Gos has also been shown to indirectly inhibit Notch (and Wnt) signaling via targeting of the translational activator and repressor Musashi-1 in colon carcinoma cell lines and nude mice xenografts [18]. Similarly, it has been shown using transgenic mice that Notch and Hedgehog signaling regulate a chemotherapy-resistant and possibly stem-like subpopulation of cells in prostate cancer that can be re-sensitized by targeted therapy [61]. Based on our findings, we propose that combined Hh/Notch inhibition strongly reduces the stemlike phenotype of GS cells, which is further evidenced by the ~15-fold reduction of stem-cell frequency in GS-5 and GS-8 and 7~-fold reduction in 17/02. Our ex vivo experiments provide further support for the possible clinical potential of the ATO/Gos combination in limiting tumor growth in an organotypic environment. Moreover, (1) ATO is already used in the clinic to treat APL, (2) is tested in 5 clinical trials for glioma treatment (clinicaltrial.gov; NCT00095771; NCT00720564; NCT00045565; NCT00185861 and NCT00275067) and (3) is known to cross the blood-brain-barrier [62,63]. Gos has already been tested in a clinical trial with measurable, but not significant effects [64], which might be due to its low penetration of the blood brain barrier [65]. Future research should therefore address if the ATO/Gos combination can be applied to murine models and/or if Gos can be substituted with other cell-death inducing drugs that can better permeate the blood-brain barrier. 

In summary, it seems likely that ATO evokes a multifactorial cellular response consisting of stemness-specific (i.e., targeting the expression of stemness marker genes and inhibition of driver pathways like Hh and Notch) and general cell death-inducing (i.e., enhanced DNA damage through ROS-induction) mechanisms that are further enhanced and/or complemented by Gos (e.g., induction of mitochondrial damage), finally leading to the effective depletion of GSCs. Importantly, we could show that ATO (± Gos) leads to an almost complete abrogation of sphere forming capacity, indicating grossly reduced self-renewal capacity of GSCs. In conclusion, the combination of ATO and Gos has the potential to specifically target brain tumor stem cell populations and should be further investigated, especially in regard to the multitude of cancer hallmarks that are addressed by this treatment.

## 4. Materials and Methods 

### 4.1. Cells and Cell Culture

The glioma stem-like cells (GS) with a restricted phenotype (GSr) GS-1 and with the full stem-like phenotype (GSf) GS-3, GS-5 and GS-8 were a kind gift from Kathrin Lamszus (UKE Hamburg, Germany) and have been described previously [66]. GS-5 and GS-8 were transduced with a construct for the stable expression of GFP/Luciferase as described previously [67]. The primary GS-lines were prepared as described previously [68]. 17/01 is from a 51 year old female, 17/02 is from a recurrent tumor of a 60 year old male. The initial tumor was treated with percutaneous radiotherapy (60 Gy) followed by four cycles of adjuvant TMZ chemotherapy followed by further cycles of TMZ therapy with reduced dosages. The recurrent tumor was first treated with fractionated irradiation (36 Gy) and subtotally resected. From this resection, the line was established. Both tumors were classified as GBM (IDH1-neg.). Tumor samples were obtained after patients gave informed consent. The University Cancer Center (UCT; Universitäres Centrum für Tumorkrankheiten Frankfurt) of the University Hospital Frankfurt provided the biomaterial after approval of the local ethics committee (SNO-12-2016). An STR profile was generated for all GS-cell lines using the multiplex STR system for human identification Power Plex 21^®^ from Promega (Mannheim, Germany) performed by Genolytic GmbH (Leipzig, Germany) in 2018, except GS-8 GFP-Luc. All cells display a different genotype and were positively compared to DNA from the original tumor or from very-low passage cells (kindly provided by K. Lamszus (Department of Neurological Surgery, UKE Hamburg, Germany) except GS-5 GFP-Luc, which is derived from GS-5 and harbors the same genotype as GS-5. All GS-cells were kept in Neurobasal A Medium (Gibco, Darmstadt, Germany) containing B27 Supplement (Gibco), 100 U/mL Penicillin 100 µg/mL Streptomycin (Gibco), 1× Glutamax (Gibco), 1× B27 Supplement (Gibco) and 20 ng/mL epidermal growth factor (EGF, Peprotech, Hamburg, Germany) and fibroblast growth factor (FGF, Peprotech). HEK-293-T, HEK293-Shh and Shh light II were cultured as described [22]. The GS-cells were dissociated using Accutase (Sigma-Aldrich, Taufkirchen, Germany); all other cells using Trypsin/EDTA (Gibco) to create a single cell suspension prior to seeding. All cells are tested monthly for mycoplasma using the PCR Mycoplasma Test Kit II (AppliChem, Darmstadt, Germany) according to the manufacturer’s instructions.

### 4.2. Compounds

Arsenic trioxide (As_2_O_3_, ATO; Sigma-Aldrich) was solved in 1 M NaOH, diluted with PBS (Gibco) to 0.5 M and solved at 80 °C while stirring. The solution was than sterile filtrated and diluted to 1 mM for long-term storage. An intermediate dilution of 15 mM was used for all consecutive dilutions. AT-101 ((−)-Gossypol, Bio-Techne GmbH, Wiesbaden--Nordenstadt, Germany), GANT61 (Sigma-Aldrich) and etoposide (Enzo Life Sciences, Lörrach, Germany) were diluted in DMSO (Carl Roth GmbH, Karlsruhe, Germany).

### 4.3. SDS-PAGE and Western Blot

Western Blotting was carried out as described previously [69]. After blocking with 5% BSA/TBS-Tween 20 (TBS-T) or 5% Milk/TBS-T the primary antibodies were incubated overnight in 5% BSA/TBS-T at 4 °C, while secondary goat anti-mouse, goat anti-rabbit or donkey anti-goat antibodies (dilution 1:10,000, LI-COR Biosciences, Bad Homburg, Germany) were incubated at room temperature for 1 h. Detection was achieved using a LI-COR Odyssey reader (LI-COR Biosciences).

### 4.4. Immunofluorescence Microscopy

For immunofluorescent microscopy of GS-5 10,000 to 12,000 cells were seeded on Laminin-coated 8-well chamber slides (Falcon, Corning, Amsterdam, NY, USA). Laminin-coating (10 µg/mL, sigma, L2020) was performed at 4 °C overnight. One day after seeding the cells were treated as indicated and fixed with 4% paraformaldehyde for 20 min at RT. The slides were washed with TBS-Tween (0.1%; TBS-T), blocked with 4% BSA in TBS with 0.3% Triton X-100 for 1 h at RT and primary antibody incubation occurred at 4 °C overnight. Hereafter, the slides were washed at least three times with TBS-T and secondary antibody was diluted 1:500 in TBS-T and incubated for 1 h at RT. After an additional wash with TBS-T the slides were mounted with DAPI containing Immunoselect antifading mounting medium (Dianova, Hamburg, Germany) or Fluoroshield with DAPI (Thermo Fisher, Frankfurt, Germany). Images were acquired with an Eclipse TS100 inverted fluorescence microscope (Nikon, Düsseldorf, Germany) operated by NIS Elements AR software (version 3.22, Nikon) or an AxioImager Z1 (Carl Zeiss, Jena, Germany).

### 4.5. Antibodies

The following antibodies and dilutions were used: Sox9 (#ab185966, abcam, Cambridge, UK) 1:5000 for western blot (WB); 1:500 for immunofluorescence (IF); Sox2 (#MAB2016, R&D Systems, Wiesbaden, Germany) 1:1000 (WB) and 1:250 (IF); Olig2 (#AF2418, R&D Systems) 1:10,000 (WB) and 1:5000 (IF); GAPDH (#CB1001, Calbiochem, Darmstadt, Germany) 1:20,000; CHK1 (#2360, Cell Signaling Technologies (CST), Frankfurt am Main, Germany) 1:1000; phosphoCHK1 (CST #2348), 1:1000; CHK2 (CST #2662S) 1:1000; Survivin (R&D #AF886) 1:1000; TP53BP1 (NB #100-304, Novus Biologicals, Wiesbaden, Germany) 1:1000; γH2AFXSer139 (#05-636, clone JBW301, Merck Millipore, Darmstadt, Germany) 1:1000; F(ab′)2-Goat anti-Rabbit IgG (H + L) Cross-Adsorbed Secondary Antibody, Alexa Fluor 488 (A-11070, Thermo Fisher) 1:500; F(ab′)2-Goat anti-mouse IgG (H + L) Cross-Adsorbed Secondary Antibody, Alexa Fluor 594 (A-11020, Thermo Fisher) 1:500; F(ab′)2-donkey anti-goat IgG (H + L) Cross-Adsorbed Secondary Antibody, Alexa Fluor 488 (A-11055, Thermo Fisher) 1:500; donkey anti-goat IgG (sc2042, Santa Cruz, Dallas, TX, USA) 1:10,000. 

### 4.6. Cell-Based Assays

MTT (3-(4,5-Dimethylthiazol-2-yl)-2,5-diphenyltetrazolium bromide) assay was basically performed as described previously [70] and measured on a Tecan Genios or Spark (Tecan, Grödig, Austria) plate reader at 560 nm. MTT (Sigma-Aldrich) was solved in sterile PBS at 5 mg/mL. The combination index (CI) [21] was calculated as described previously [71] using CompuSyn (combosyn.com) software [21]. Cell death was measured using Annexin-V-APC (BD Biosciences, Heidelberg, Germany) and propidium iodide (PI, Merck) as described previously [69] on a BD Accuri C6 (BD Biosciences). To measure the sphere forming ability an extended sphere formation assay (SFA) based on Gilbert et al. [33] was performed. Briefly, 6000 single cells were seeded in 6 well plates in the presence of the compound indicated and measured after 10, 20 and 30 days (for details see Figure 5C). Limiting dilution assay (LDA) was performed as described [72] by seeding serial dilutions of 4 to 500 cells/well for GS-5 GFP-Luc and GS-8 and 8 to 1024 cells/well for 17/02 and analyzed using extreme limiting dilution analysis (ELDA) software [34]. For DNA damage analyses the cells were seeded on laminin-coated (10 µg/mL, Sigma-Aldrich) 8-well-chamber-slides (Corning, Wiesbaden, Germany) and processed as described previously [73]. For GANT-functionality tests using Shh light II Sonic hedgehog conditioned medium (Shh-CM) or control medium (CoM) was prepared as described [71]. For the reporter assay 10,000 Shh light II cells were seeded in lumitrac 600 96-well plates (Greiner bio-one, Frickenhausen, Germany) in culture medium. The next day the medium was replaced with CoM or Shh-CM and after 24 h the cells were treated as indicated in Shh-CM for 48 h. Hereafter a dual-luciferase assay (Promega, Mannheim, Germany) was performed per the manufacturer’s instructions.

### 4.7. Proteomics

For mass-spectrometry-based proteomics the cells were seeded in 6-well plates at 300,000 cells per well in triplicates and treated for 24 h. Sample-preparation was performed as described recently [74] with minor modifications. In brief, a cell lysate was digested in a urea buffer with LysC and Trypsin after reduction and alkylation of disulfide bonds with DTT and iodoacedamide. Tryptic peptides were analysed on a QExactive HF after reversed phase separation on an easy nLC 1200. Data were analysed with MaxQuant and differentially expressed proteins were identified by MaxLFQ quantification [75] applying an FDR of 0.05 and an S0 of 0.2 as cut-offs. The mass spectrometry proteomics data have been deposited to the ProteomeXchange Consortium via the PRIDE [76] partner repository with the dataset identifier PXD009249.

### 4.8. Taqman-Based qRT-PCR

RNA-Isolation was conducted using the RNeasy Mini Kit (Qiagen, Hilden, Germany) in combination with the QiaShredder Kit or the ExtractMe Total RNA Kit (Blirt S.A., Gdansk, Poland) according to the manufacturer’s instruction. cDNA-Synthesis was achieved using the SuperScript III System (Life Technologies, Darmstadt, Germany) according to the manufacturers instruction using 100 U SuperScript per sample. The quantitative Real-Time PCR (qRT-PCR) was performed using 20× Taqman Probes (Applied Biosystems, Darmstadt, Germany) and 2× Fast-Start Universal Probe Master Mix (Roche, Mannheim, Germany) on a StepOne Plus System (Applied Biosystems) using the standard setting. The target gene expression values were normalized to the reference gene TATA-box binding protein (TBP). A list of all FAM-MGB probes is provided as a Appendix A.

### 4.9. Adult Organotypic Slice Cultures and Ex Vivo Tumor Growth assay

Adult, organotypic brain slices (OTC) from 8–12 week old C56BL/6JOlaHsd mice were prepared as recently described [77]. Briefly, the brains were embedded in 2% low-melting agarose (Carl Roth) and cut on a Vibratome VT1000 (Leica, Wetzlar, Germany) in 250 µm thick transverse sections and cultured on Millicell cell culture inserts (PICMORG50, Merck) in 6-well plates containing OTC-specific, FCS-free medium (DMEM/F12, 1× B27, 1× N2-supplement, 1% P/S; all from Gibco). For the ex vivo tumor growth assay 2000 GFP-Luc positive GS-5 or GS-8 or CellTracker CM Dil-labeled (Molecular Probes, Thermo Fisher, dilution: 1:1000) 17/02 were grown in u-shaped 96-well plates 4–6 days prior to OTC preparation in order to generate large spheres. One day after OTC-preparation multiple spheres were spotted onto the OTCs by careful pipetting. Starting from the next day after spotting (hereafter defined as Day 1) the OTCs were treated as indicated three times per week by exchanging the medium. Tumor growth was monitored using a Nikon SMZ25 stereomicroscope equipped with a P2-SHR Plan Apo 2× objective operated by NIS elements software (version 4.30.02). The tumor area was measured and processed using FIJI [78]. To generate growth curves each tumor was normalized to the size of day 1. All animal experiments comply with the ARRIVE guidelines and were performed in accordance with the German animal protection law authorized by the regional administrative council (Regierungspräsidium Darmstadt, Germany) to the department of medicine (University Hospital Frankfurt).

### 4.10. Statistics

All statistical analyses applying One-ANOVA followed by Tukey Post-Hoc-Tests were performed using GraphPad Prism 7 (GraphPad Software, La Jolla, CA, USA) basically as described previously [71].

## 5. Conclusions

Glioblastoma recurrence is in part mediated by so-called glioblastoma stem-like cells (GSCs) that are characterized by expression of stemness marker genes, high Hedgehog and Notch pathway activity and treatment resistance through enhanced DNA repair. Here, we show for the first time that arsenic trioxide-mediated Hedgehog/Notch inhibition in combination with the natural anticancer agent (−)-Gossypol synergistically targets GSCs in part by interfering with DNA double strand break repair by reducing CHEK1 and CHEK2 expression. The combination of ATO and Gos was also proven successful in ex vivo models where it could strongly reduce tumor growth and even eliminate some tumors.

## Figures and Tables

**Figure 1 cancers-11-00350-f001:**
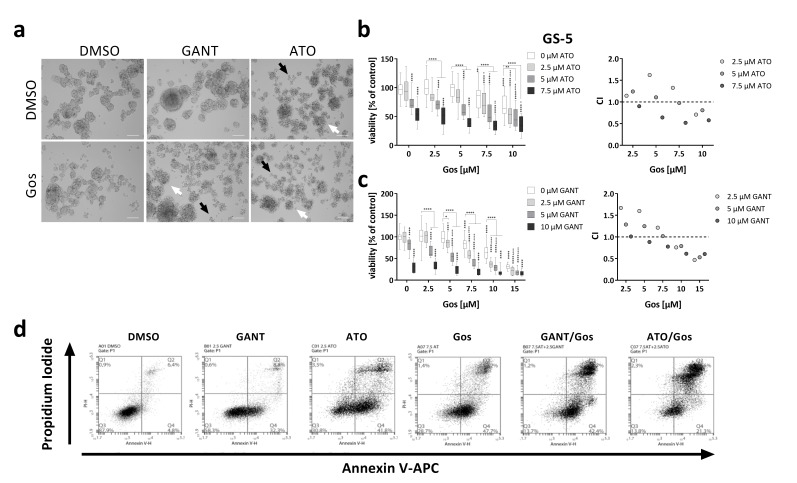
Synergistic inhibition of viability and induction of cell death of GS-5 and 17/02 GSCs via inhibition of Hh and Notch signaling. (**a**) Morphological appearance of GS-5 24 h after treatment with solvent (DMSO), 5 µM GANT, 5 µM ATO or 7.5 µM Gos alone or in combination. Note that after ATO (upper right) single treatment more single cells (black arrows) and disintegrating spheres (white arrows) can be detected which is further enhanced by ATO/Gos combination treatment and also after GANT/Gos; scale bar: 100 µm. (**b**,**c**, left side) Box-Plots (Tukey) of MTT assay of GS-5 after treatment for 24 h with increasing concentrations of (**b**) ATO or (**c**) GANT in combination with Gos. (**b**,**c**, right side) The CI was calculated from the data obtained according to Chou et al. [21] using the non-constant ratio setting (CI > 1: antagonism; CI = 1: additive; CI < 1: synergism). The CI value is given as a single value calculated from the summary of all experiments. (**d**) Representative Dot-Plots of GS-5 after treatment for 24 h with solvent (DMSO), 2.5 µM GANT, 2.5 µM ATO or 7.5 µM Gos alone or in combination. (**e**) Stacked bar chart of GS-5 after treatment for 24 h with the drugs and concentrations as indicated. (**f**) Box-Plots (Tukey) of the percentages of dead cells (100%—An-/PI-) after treatment. The lines in the Box-Plots represents the median, the plus-symbol the mean. (**g**,**h**) Bar graph of Taqman-based gene expression analysis of (**g**) GS-5 and (**h**) 17/02 after treatment with 5 µM Gos, 2.5 µM GANT or ATO or the combination of GANT or ATO with Gos (G + Gos; A + Gos). The MTT assays were performed at least three times in 6 biological replicates. All other experiments were performed at least three times in three biological replicates. Error bars are SEM. * *p* < 0.05; ** *p* < 0.01; *** *p* < 0.001; **** *p* < 0.0001 against solvent or as indicated; ° *p* < 0.05; °° *p* < 0.01; °°° *p* < 0.001; °°°° *p* < 0.0001 against GANT or ATO single treatment; # *p* < 0.05 against both single treatments.

**Figure 2 cancers-11-00350-f002:**
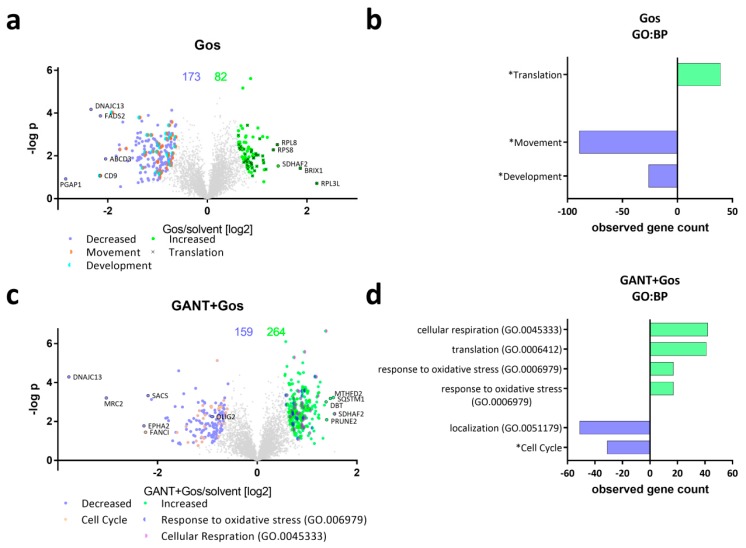
GANT and ATO-treatment in combination with Gos causes global proteomic changes affecting differentiation, cell cycle and proliferation. (**a**,**c**,**e**) Volcano Plots showing the protein ratios (in log2) as a function of the –log p-values of label-free quantification proteomics data of GS-5 after treatment for 24 h with (**a**) 5 µM Gos alone or in combination with (**c**) 2.5 µM GANT or (**e**) 2.5 µM ATO. 5008 proteins were quantified, the amount of significantly changed proteins is written above the Plot. (**b**,**d**,**f**) Bar Chart after bioinformatic analysis using the STRING-platform (string-db.com, [26]) and Revigo (revigo.irb.hr, [27]) for enriched Gene Ontology biological processes (GO:BP) and manual clustering (marked with an *) after treatment with (**b**) Gos alone, (**d**) GANT + Gos or (**f**) ATO + Gos. (**g**) Venn diagram depicting the overlap between the significantly (left side) decreased and (right side) increased proteins. Note that the x-axis in (**b**,**d**,**f**) depicts the number of proteins that are significantly decreased (negative values) and increased (positive values).

**Figure 3 cancers-11-00350-f003:**
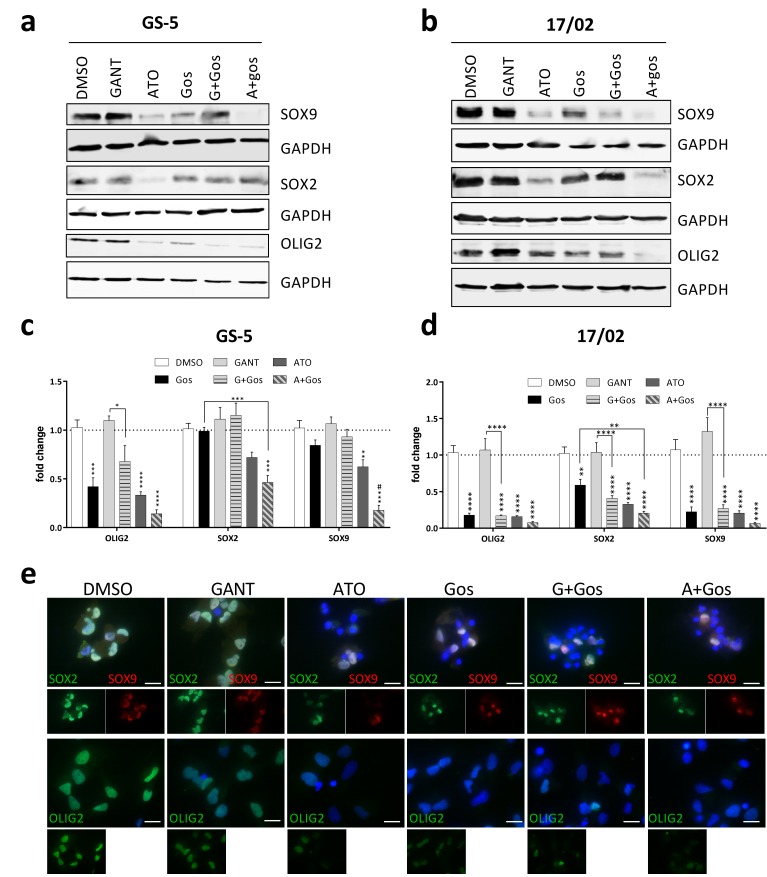
ATO, but not GANT in combination with Gos reduces GSC marker and Hh and Notch pathway activity. (**a**,**b**) Western Blot analysis of (**a**) GS-5 and (**b**) 17/02 after treatment for 24 h with solvent (DMSO), 7.5 µM Gos, 5 µM GANT, 5 µM ATO or a combination of Gos with GANT (GANT + Gos) or ATO (ATO + Gos). (**c**,**d**) Bar graph of Taqman-based gene expression analysis of GS-5 cells after treatment with 5 µM Gos, 2.5 µM GANT or ATO or the combination of GANT or ATO with Gos (G+Gos; A+Gos). (**e**) Representative images of immunofluorescent microscopy of GS-5 72h after treatment as in (**c**) of the stemness markers SOX2 (green) and SOX9 (red) (upper row) and OLIG2 (green; lower row); scale bar: 20 µm. (**f**) Summarized quantification of three independent experiments; at least 100 cells per condition and experiment were counted. All experiments were performed at least 3 times; gene expression data was further performed in 3 biological replicates. * *p* < 0.05; ** *p* < 0.01; *** *p* < 0.001; **** *p* < 0.0001. # *p* < 0.05; ## *p* < 0.01; ### *p* < 0.001; #### *p* < 0.0001. against both single treatments One-way ANOVA followed by Tukey Post-Hoc-Test (GraphPad Prism 7).

**Figure 4 cancers-11-00350-f004:**
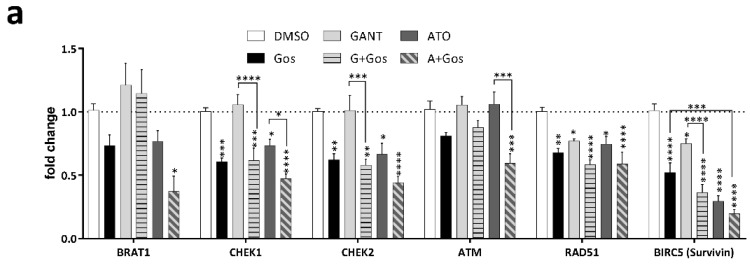
GANT or ATO in combination with Gos synergistically induces DNA damage in GSC via downregulation of CHK1, CHK2 and Survivin. (**a**) Taqman-based gene expression analysis of GS-5 after treatment with 5 µM Gos, 2.5 µM GANT or ATO or the combination of GANT or ATO with Gos (G + Gos; A + Gos). (**b**) Western Blot analysis of GS-5 after treatment for 24 h with solvent (DMSO), 7.5 µM Gos, 5 µM GANT, 5 µM ATO or a combination of Gos with GANT (GANT + Gos) or ATO (ATO + Gos). (**c**,**d**) Dot-Plots of (**c**) TP53BP1 (green)- or (**d**) γH2AFX (red)-positive foci per nucleus of GS-5 24 h after treatment with 5 µM Gos, 3 µM GANT, 2.5 µM ATO or a combination of GANT and Gos (GANT + Gos) or ATO and Gos (ATO + Gos). Each point represents the number of foci per nucleus. Representative pictures are presented in (**e**); scale bar: 20 µm. The experiments were performed at least 3 times in triplicates. For each replicate at least 20 nuclei were counted. Statistics in (**c**,**d**) were performed using the mean value of each replicate. * *p* < 0.05; ** *p* < 0.01; *** *p* < 0.001; **** *p* < 0.0001 against solvent; # *p* < 0.05 against both single treatments. One-way ANOVA followed by Tukey Post-Hoc-Test (GraphPad Prism 7).

**Figure 5 cancers-11-00350-f005:**
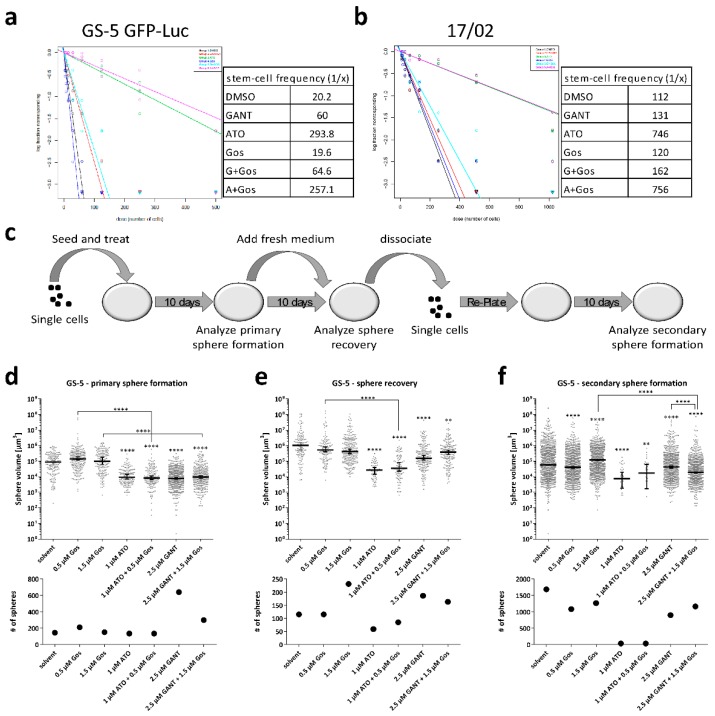
ATO and Gos treatment reduces stemness properties of GS-5/GS-5 GFP-Luc and 17/02 in vitro. (**a**,**b**) Extreme limiting dilution analysis [34] of (**a**) GS-5 GFP-Luc and (**b**) 17/02 7 days after seeding and treatment with 1 µM ATO, 2.5 µM GANT61 or 0.5 µM Gos alone or in combinaton of increasing cell numbers (4 to 500 cells for GS-5 and 8 to 1024 for 17/02). The stem cell frequency calculated by ELDA software [34] is depicted next to the graph. (**c**) Scheme of the extended SFA. (**d**–**f**) Dot-Plots displaying the (upper row) sphere volume and (lower row) total number of spheres of GS-5 spheres after measurement of primary sphere formation, sphere recovery and secondary sphere formation. Lines in (**d**–**f**) are the median ± 95% confidence intervals. Limiting dilution assays were performed three times in 12 biological replicates. Sphere formation assays were performed twice in triplicates and three vision fields were analyzed for each biological replicate. * *p* < 0.05; ** *p* < 0.01; *** *p* < 0.001; **** *p* < 0.0001. One-way ANOVA followed by Tukey Post-Hoc-Test (GraphPad Prism 7).

**Figure 6 cancers-11-00350-f006:**
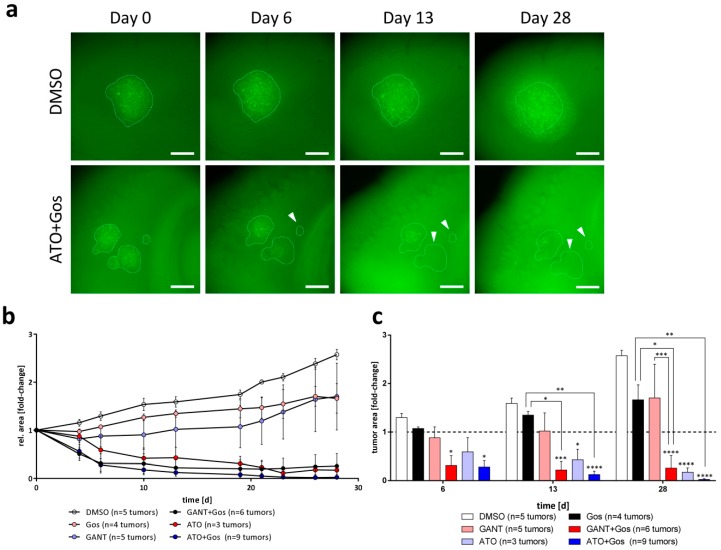
ATO, ATO/Gos and GANT/Gos decrease tumor size in adult OTC transplantation experiments. (**a**) Representative images of tumors of GS-5 GFP-Luc grown on adult OTCs after treatment with solvent (DMSO) or ATO/Gos. Within 6 days some tumors treated with ATO/Gos show complete elimination (white arrows), while others decrease in size; scale bar: 500 µm. (**b**) Growth kinetic of tumors over time after treatment with solvent (DMSO, white), 2.5 µM GANT (pink), 2.5 µM ATO (light blue), 5 µM Gos (black), GANT/Gos (red) or ATO/Gos (blue). (**c**) Bar graph for selected time point displaying the mean (+ SEM) tumor size. * *p* < 0.05; ** *p* < 0.01; *** *p* < 0.001; **** *p* < 0.0001 compared to DMSO for each time-point. One-way ANOVA followed by Tukey Post-Hoc-Test (GraphPad Prism 7).

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
