# Peer review of "Arsenic Trioxide and (−)-Gossypol Synergistically Target Glioma Stem-Like Cells via Inhibition of Hedgehog and Notch Signaling"

_cancers, 2019, doi:10.3390/cancers11030350_

Round 1
Reviewer 1 Report
1. The study would benefit from the use of a biochemical and/or genetic approach to inhibit Notch pathway in addition to Hh, demonstrating that Notch and Hh inhibition is indeed critical for the enhanced response to Gos. 2. ATO is known to affect multiple signaling pathways (e.g. via thiol adduction, ROS generation), and its effects are not limited to Hh and Notch. To what extent to these other actions of ATO contribute to altering the stemness of glioblastoma cell lines, and to what extent might they enhance the response to Gos through stemness-independent processes? 3. The main hypothesis of the study (lines 73-74) is that the enforced differentiation of glioblastoma would make them more vulnerable to cell death (by Gos). The inhibitory effects of ATO alone on stemness markers in GS-5 cells is demonstrated clearly in Fig 5a, but not in proteomic data (supplemental file 2), which should be explained. 4. Gos alone appears to have inhibitory effects on stemness marker gene and protein expression, as well as on Hh and Notch signaling. These contribution cannot be ruled out when looking at the combination of Gos + ATO, and significantly confuses the interpretation of the data.
Author Response
1. The study would benefit from the use of a biochemical and/or genetic approach to inhibit Notch pathway in addition to Hh, demonstrating that Notch and Hh inhibition is indeed critical for the enhanced response to Gos.
Answer: We assume the reviewer suggests to perform single inhibition of the Notch pathway in combination with Gos treatment. We agree that this would be a nice addition to our approach, however our intention was rather to compare single vs dual pathway inhibition. We chose two pathways that are known to interact (Kong et al. 2015); are known to regulate the stem-like phenotype of cancer stem cells including GSC (Clement et al. 2007; Altaba 2011; Hsieh et al. 2011; Floyd et al. 2012; Purow 2012; Ding et al. 2014; Yahyanejad et al. 2016) and can be targeted using small molecules (Beauchamp et al. 2011; Wu et al. 2013; Ding et al. 2014). We therefore decided to focus on inhibition of only one single (Hh) and the dual pathways. Additionally, did we focus on Hh signaling, because to our knowledge are the available inhibitors for Hh signaling (e.g. GANT61) more specific compared to Notch-inhibitors. In fact, pharmacological Notch inhibition is usually achieved using gamma-secretase inhibitors. But since this enzyme is also involved in a variety of other processes like APP-processing (Siemers et al. 2006), MHC-1 cleavage (Carey et al. 2007) and assembly of adherens junctions (Marambaud et al. 2002) we preferred the Hh inhibitor.
As for the genetic approach: We explicitly decided to focus on easily translatable results with small-molecule inhibitors, because genetic approaches (at least of single pathway inhibition) have been used extensively by others (Clement et al. 2007; Xu et al. 2008; Fan et al. 2010; Takezaki et al. 2011; Vermezovic et al. 2015). Therefore, we applied clinically applicable drugs that target these pathways and have emphasized the rationale for using small molecule inhibitors now. Although we agree that a genetic validation of our working hypothesis would mechanistically enhance our findings, this would be beyond the scope of a single manuscript given the wealth of information we are already presenting.
2. ATO is known to affect multiple signaling pathways (e.g. via thiol adduction, ROS generation), and its effects are not limited to Hh and Notch. To what extent to these other actions of ATO contribute to altering the stemness of glioblastoma cell lines, and to what extent might they enhance the response to Gos through stemness-independent processes?
Anwer: This is an excellent point. We agree with the reviewer that the response of the cells to ATO is very likely a multi-factorial response including stemness-independent processes and we regret if we have not discussed this in due detail. In fact we believe that one of the reasons ATO is so effective is because it affects multiple processes simultaneously, processes that might very well interact with each other. Of course this pleiotropic mode-of-action is a double-edged sword because it makes it very hard to pinpoint the molecular target to a single protein. We have now thoroughly re-checked our proteomic dataset and the literature and have further discussed our findings especially focusing on previously unmentioned processes (lines 420 to 443). But to discuss all potential modes of action of ATO and/or ATO+Gos would be beyond the scope of this manuscript. Hence we only focused on the most obvious processes (DNA damage/ROS-production) and the covalent binding to proteins (Thiol-adduction) as suggested by the reviewer.
3. The main hypothesis of the study (lines 73-74) is that the enforced differentiation of glioblastoma would make them more vulnerable to cell death (by Gos). The inhibitory effects of ATO alone on stemness markers in GS-5 cells is demonstrated clearly in Fig 5a, but not in proteomic data (supplemental file 2), which should be explained.
Answer: We would like to thank the reviewer for pointing this out. We agree that the effects on stemness are not visible in the data shown in the proteomic dataset, which is likely due to the relatively low number of significantly changed proteins. First of all is it important to note that we chose rather stringent analysis parameters for the proteomic dataset (FDR: 0.05 and an S0-value (FYI: minimal fold change) of 0.2). We (Das et al. 2018; Meyer et al. 2018) and others (Bullinger et al. 2013; Jiang et al. 2017) applied these methods previously, whereas simpler approach solely based on p-value cut-offs can also be found in the literature (Pavelek et al. 2016; Galdiero et al. 2018). Application of this simpler approach leads to vastly higher numbers of significantly regulated proteins. Of course the question arises whether more proteins is always better, especially if the changes compared to solvent-treated cells are quite low (or if inter-replicate variation is quite high; resulting in lower p-values. To exemplify this, we analyzed the proteomic data of GANT61(which showed only one significantly regulated protein according to our stringent analysis), ATO and ATO/Gos-treated GS-5 again and applied only the p-value as a cut-off (Fig. 1 for response only).
As can be clearly seen is the number of differentially regulated proteins vastly increased if only a p-value approach is taken. In fact, using this cut-off we can now show that the three stemness marker OLIG2, SOX2 and SOX9 are now “significantly” decreased. However, for omic-analyses we prefer to “loose” some data points in order to improve the biological relevance.
[please see uploaded version]
Figure 1 (For response only): Different cut-offs for proteomic data. This figure illustrates how the current analysis of the proteomic data-set using an FDR of 0.05 and an S0 of 0.2 (A to C) differs from a simpler analysis using only p-value (D to F). This is shown for GANT alone (A and D; because GANT alone only induces significant changes of one protein according to the stringent analysis), ATO (B and E) and ATO+Gos (C and F). For the sake of completeness, p-values<0.01 are also depicted in (D to F). Note that even with a more stringent p-value far more proteins would have been marked as significantly regulated although the absolute differences (i.e. changes from X=0) are quite small. Also note that ATO regulates the three stemness marker OLIG2, SOX2 and SOX9 using the p-value-based analysis.
Additionally, would we like to point out that we used lower concentrations for the proteomic approach, than for example for the Western Blots, where ATO clearly reduces stemness related marker proteins. As mentioned in the Results section did we use lower concentrations of ATO (i.e. 2.5 µM), because the higher (5 µM) concentrations (this is especially the case for the combination treatments) might have already resulted in excessive cell death and this would have masked many of the underlying processes. This is of course the case for all drugs and not only ATO. Finally, would like to point out that changes in protein abundance are usually preceded by changes in RNA abundance. Accordingly, we can show in multiple GS-lines and primary cells that ATO (2.5 µM) effectively inhibits mRNA-expression of the three stemness markers. Nonetheless do we agree that this apparent contradiction required further explanation and we have added a clarification in the discussion section (lines 407 to 412).
4. Gos alone appears to have inhibitory effects on stemness marker gene and protein expression, as well as on Hh and Notch signaling. These contribution cannot be ruled out when looking at the combination of Gos + ATO, and significantly confuses the interpretation of the data.
Answer: The reviewer is correct that we should have discussed these findings in more detail. We have carefully examined the existing literature and have discussed our findings in regard to the effects of Gos on the mentioned signaling pathways. Nonetheless we are confident that our hypothesis is strongly supported by our data presented in figures 5 and 6 (and Fig S7 to S9) showing that ATO is the only of the three drugs that exhibits pronounced phenotypic changes indicating reduced stemness and also strongly reduces tumor growth in ex vivo assays. We have now extended our discussion on the effects of Gos (lines 412 to 419).
Submission Date
28 December 2018
Date of this review
16 Jan 2019 22:59:28
References:
Altaba, A. R. I. (2011). "Hedgehog Signaling and the Gli Code in Stem Cells, Cancer, and Metastases." Science Signaling 4(200).
Beauchamp, E. M., Ringer, L., Bulut, G., Sajwan, K. P., Hall, M. D., Lee, Y. C., Peaceman, D., Ozdemirli, M., Rodriguez, O., Macdonald, T. J., Albanese, C., Toretsky, J. A. and Uren, A. (2011). "Arsenic trioxide inhibits human cancer cell growth and tumor development in mice by blocking Hedgehog/GLI pathway." The Journal of clinical investigation 121(1): 148-160.
Bullinger, L., Schlenk, R. F., Gotz, M., Botzenhardt, U., Hofmann, S., Russ, A. C., Babiak, A., Zhang, L., Schneider, V., Dohner, K., Schmitt, M., Dohner, H. and Greiner, J. (2013). "PRAME-induced inhibition of retinoic acid receptor signaling-mediated differentiation--a possible target for ATRA response in AML without t(15;17)." Clinical cancer research : an official journal of the American Association for Cancer Research 19(9): 2562-2571.
Carey, B. W., Kim, D. Y. and Kovacs, D. M. (2007). "Presenilin/gamma-secretase and alpha-secretase-like peptidases cleave human MHC Class I proteins." Biochem J 401(1): 121-127.
Clement, V., Sanchez, P., de Tribolet, N., Radovanovic, I. and Ruiz i Altaba, A. (2007). "HEDGEHOG-GLI1 signaling regulates human glioma growth, cancer stem cell self-renewal, and tumorigenicity." Current biology : CB 17(2): 165-172.
Das, C. K., Linder, B., Bonn, F., Rothweiler, F., Dikic, I., Michaelis, M., Cinatl, J., Mandal, M. and Kogel, D. (2018). "BAG3 Overexpression and Cytoprotective Autophagy Mediate Apoptosis Resistance in Chemoresistant Breast Cancer Cells." Neoplasia 20(3): 263-279.
Ding, D., Lim, K. S. and Eberhart, C. G. (2014). "Arsenic trioxide inhibits Hedgehog, Notch and stem cell properties in glioblastoma neurospheres." Acta neuropathologica communications 2: 31.
Fan, X., Khaki, L., Zhu, T. S., Soules, M. E., Talsma, C. E., Gul, N., Koh, C., Zhang, J., Li, Y. M., Maciaczyk, J., Nikkhah, G., Dimeco, F., Piccirillo, S., Vescovi, A. L. and Eberhart, C. G. (2010). "NOTCH pathway blockade depletes CD133-positive glioblastoma cells and inhibits growth of tumor neurospheres and xenografts." Stem Cells 28(1): 5-16.
Floyd, D. H., Kefas, B., Seleverstov, O., Mykhaylyk, O., Dominguez, C., Comeau, L., Plank, C. and Purow, B. (2012). "Alpha-secretase inhibition reduces human glioblastoma stem cell growth in vitro and in vivo by inhibiting Notch." Neuro-oncology 14(10): 1215-1226.
Galdiero, F., Bello, A. M., Spina, A., Capiluongo, A., Liuu, S., De Marco, M., Rosati, A., Capunzo, M., Napolitano, M., Vuttariello, E., Monaco, M., Califano, D., Turco, M. C., Chiappetta, G., Vinh, J. and Chiappetta, G. (2018). "Identification of BAG3 target proteins in anaplastic thyroid cancer cells by proteomic analysis." Oncotarget 9(8): 8016-8026.
Hsieh, A., Ellsworth, R. and Hsieh, D. (2011). "Hedgehog/GLI1 regulates IGF dependent malignant behaviors in glioma stem cells." Journal of cellular physiology 226(4): 1118-1127.
Jiang, X., Feng, L., Dai, B., Li, L. and Lu, W. (2017). "Identification of key genes involved in nasopharyngeal carcinoma." Braz J Otorhinolaryngol 83(6): 670-676.
Kong, J. H., Yang, L., Dessaud, E., Chuang, K., Moore, D. M., Rohatgi, R., Briscoe, J. and Novitch, B. G. (2015). "Notch activity modulates the responsiveness of neural progenitors to sonic hedgehog signaling." Developmental cell 33(4): 373-387.
Marambaud, P., Shioi, J., Serban, G., Georgakopoulos, A., Sarner, S., Nagy, V., Baki, L., Wen, P., Efthimiopoulos, S., Shao, Z., Wisniewski, T. and Robakis, N. K. (2002). "A presenilin-1/gamma-secretase cleavage releases the E-cadherin intracellular domain and regulates disassembly of adherens junctions." Embo Journal 21(8): 1948-1956.
Meyer, N., Zielke, S., Michaelis, J. B., Linder, B., Warnsmann, V., Rakel, S., Osiewacz, H. D., Fulda, S., Mittelbronn, M., Munch, C., Behrends, C. and Kogel, D. (2018). "AT 101 induces early mitochondrial dysfunction and HMOX1 (heme oxygenase 1) to trigger mitophagic cell death in glioma cells." Autophagy 14(10): 1693-1709.
Pavelek, Z., Vysata, O., Tambor, V., Pimkova, K., Vu, D. L., Kuca, K., Stourac, P. and Valis, M. (2016). "Proteomic analysis of cerebrospinal fluid for relapsing-remitting multiple sclerosis and clinically isolated syndrome." Biomed Rep 5(1): 35-40.
Purow, B. (2012). "Notch inhibition as a promising new approach to cancer therapy." Advances in experimental medicine and biology 727: 305-319.
Siemers, E. R., Quinn, J. F., Kaye, J., Farlow, M. R., Porsteinsson, A., Tariot, P., Zoulnouni, P., Galvin, J. E., Holtzman, D. M., Knopman, D. S., Satterwhite, J., Gonzales, C., Dean, R. A. and May, P. C. (2006). "Effects of a gamma-secretase inhibitor in a randomized study of patients with Alzheimer disease." Neurology 66(4): 602-604.
Takezaki, T., Hide, T., Takanaga, H., Nakamura, H., Kuratsu, J. and Kondo, T. (2011). "Essential role of the Hedgehog signaling pathway in human glioma-initiating cells." Cancer science 102(7): 1306-1312.
Vermezovic, J., Adamowicz, M., Santarpia, L., Rustighi, A., Forcato, M., Lucano, C., Massimiliano, L., Costanzo, V., Bicciato, S., Del Sal, G. and d'Adda di Fagagna, F. (2015). "Notch is a direct negative regulator of the DNA-damage response." Nature structural & molecular biology 22(5): 417-424.
Wu, J., Ji, Z., Liu, H., Liu, Y., Han, D., Shi, C., Wang, C., Yang, G., Chen, X., Shen, C., Li, H., Bi, Y., Zhang, D. and Zhao, S. (2013). "Arsenic trioxide depletes cancer stem-like cells and inhibits repopulation of neurosphere derived from glioblastoma by downregulation of Notch pathway." Toxicology letters 220(1): 61-69.
Xu, Q., Yuan, X., Liu, G., Black, K. L. and Yu, J. S. (2008). "Hedgehog signaling regulates brain tumor-initiating cell proliferation and portends shorter survival for patients with PTEN-coexpressing glioblastomas." Stem Cells 26(12): 3018-3026.
Yahyanejad, S., King, H., Iglesias, V. S., Granton, P. V., Barbeau, L. M., van Hoof, S. J., Groot, A. J., Habets, R., Prickaerts, J., Chalmers, A. J., Eekers, D. B., Theys, J., Short, S. C., Verhaegen, F. and Vooijs, M. (2016). "NOTCH blockade combined with radiation therapy and temozolomide prolongs survival of orthotopic glioblastoma." Oncotarget 7(27): 41251-41264.
Reviewer 2 Report
The article details the ability of ATO and GOS to synergistically target and inhibit the HH and notch signaling. The article shows a wealth of information and is well written. Few minor issues needs to be addressed.
Although the authors used organotypic brain slice to grow tumors and treated them with the compounds, in vivo mice experiments might be needed to show the effectiveness of these compounds and the side effects.
Figure 2 all the volcano plots are not legible, the authors might change (shorten) the scales on both axis to make the plot legible.
Figure 3 and 4 the authors need to include the color legend for te microscopic pictures in the figure legends.
Supplementary figure S7A is not legible.
Author Response
The article details the ability of ATO and GOS to synergistically target and inhibit the HH and notch signaling. The article shows a wealth of information and is well written. Few minor issues needs to be addressed.
Although the authors used organotypic brain slice to grow tumors and treated them with the compounds, in vivo mice experiments might be needed to show the effectiveness of these compounds and the side effects.
Answer: We agree that the final proof of the efficacy of our drug combination would be to perform in vivo experiments. We would like to point out that there are very complex, time-consuming regularities in Germany, so it is not possible to perform these animal experiments in a reasonable amount of time. Given the wealth of data already presented in the manuscript, we chose to perform these experiments in follow-up studies in the future. Regarding side effects, there are a number of published studies with ATO and Gos claiming no observable side effects (Beauchamp et al. 2011; Kim et al. 2013; Lan et al. 2015). In fact, ATO is already used in the clinic to treat APL (Alimoghaddam 2014) so its side effects are known. Gos has also been used in humans as a male contraceptive agent (Coutinho 2002) and as a trial substance for treating gliomas (Bushunow et al. 1999) and has therefore also known side effects. Of course, a combination elicits other side effects and these need to be addressed in due detail.
Figure 2 all the volcano plots are not legible, the authors might change (shorten) the scales on both axis to make the plot legible.
Answer: Although we consider it easier to make comparisons if related parts of the figures have the same scaling, the reviewer is correct in stating that improving the legibility of this Figure is warranted. We agree that the old Figures containing the volcano plots were very condensed. We have modified Figs. 2 and S4 and are confident that the new figures are now more legible. We would like to thank the reviewer for pointing this out.
Figure 3 and 4 the authors need to include the color legend for te microscopic pictures in the figure legends.
Answer: We would like to thank the reviewer for pointing this out and we have added this information now
Supplementary figure S7A is not legible.
Answer: We would like to thank the reviewer again for the careful examination of the manuscript and we have now split Fig S7A into Figure S7 and the rest into Figure S8. We are confident that this vastly improves the readability of both figures.
Submission Date
28 December 2018
Date of this review
24 Jan 2019 21:51:30
References:
Alimoghaddam, K. (2014). "A review of arsenic trioxide and acute promyelocytic leukemia." Int J Hematol Oncol Stem Cell Res 8(3): 44-54.
Beauchamp, E. M., Ringer, L., Bulut, G., Sajwan, K. P., Hall, M. D., Lee, Y. C., Peaceman, D., Ozdemirli, M., Rodriguez, O., Macdonald, T. J., Albanese, C., Toretsky, J. A. and Uren, A. (2011). "Arsenic trioxide inhibits human cancer cell growth and tumor development in mice by blocking Hedgehog/GLI pathway." The Journal of clinical investigation 121(1): 148-160.
Bushunow, P., Reidenberg, M. M., Wasenko, J., Winfield, J., Lorenzo, B., Lemke, S., Himpler, B., Corona, R. and Coyle, T. (1999). "Gossypol treatment of recurrent adult malignant gliomas." Journal of neuro-oncology 43(1): 79-86.
Coutinho, E. M. (2002). "Gossypol: a contraceptive for men." Contraception 65(4): 259-263.
Kim, J., Aftab, B. T., Tang, J. Y., Kim, D., Lee, A. H., Rezaee, M., Chen, B., King, E. M., Borodovsky, A., Riggins, G. J., Epstein, E. H., Jr., Beachy, P. A. and Rudin, C. M. (2013). "Itraconazole and arsenic trioxide inhibit Hedgehog pathway activation and tumor growth associated with acquired resistance to smoothened antagonists." Cancer cell 23(1): 23-34.
Lan, L., Appelman, C., Smith, A. R., Yu, J., Larsen, S., Marquez, R. T., Liu, H., Wu, X., Gao, P., Roy, A., Anbanandam, A., Gowthaman, R., Karanicolas, J., De Guzman, R. N., Rogers, S., Aube, J., Ji, M., Cohen, R. S., Neufeld, K. L. and Xu, L. (2015). "Natural product (-)-gossypol inhibits colon cancer cell growth by targeting RNA-binding protein Musashi-1." Mol Oncol 9(7): 1406-1420.
Reviewer 3 Report
There are still concerns with this manuscript.
A mouse in vivo model with the primary endpoint survival would be appreciated.
Rescue experiments from the cell death are missing for the most part.
How does the combination really work? This is completely unexplored.
However, the data quality and the presentation of the experiments is fine and I feel that the data as presented is still informative even without fulfilling these issues, which would have raised the impact significantly.
Author Response
There are still concerns with this manuscript.
A mouse in vivo model with the primary endpoint survival would be appreciated.
Answer: We agree that the final proof of the efficacy of our drug combination would be to perform in vivo experiments. We would like to point out that there are very complex, time-consuming regularities in Germany, so it is not possible to perform these animal experiments in a reasonable amount of time. Given the wealth of data already presented in the manuscript, we chose to perform these experiments in follow-up studies in the future. Regarding side effects, there are a number of published studies with ATO and Gos claiming no observable side effects (Beauchamp et al. 2011; Kim et al. 2013; Lan et al. 2015). In fact, ATO is already used in the clinic to treat APL (Alimoghaddam 2014) so its side effects are known. Gos has also been used in humans as a male contraceptive agent (Coutinho 2002) and as a trial substance for treating gliomas (Bushunow et al. 1999) and has therefore also known side effects. Of course, a combination elicits other side effects and these need to be addressed in due detail.
Rescue experiments from the cell death are missing for the most part.
Answer: We assume that the Reviewer refers to a rescue from different modes of cell death (apoptosis, autophagic cell death). It is correct that we have not done these experiments because the determination of the drug-induced cell death mode was not the primary aim of this study. However, based on findings published in the literature and our own previous studies in GBM cells, we assume that excessive autophagy induction might play a role in cell killing. This hypothesis is based on our previous study showing induction of autophagic cell death (ACD) using Gos (= AT-101) in non-GSC cells (Voss et al. 2010). Similarly, ATO has been shown to induce ACD in non-GSC glioma cells as well (Kanzawa et al. 2005). A detailed determination of the mode of cell death mediated by ATO +/- Gos in GSCs would be beyond the scope of this manuscript, but we have further discussed these hypothesis.
How does the combination really work? This is completely unexplored.
Answer: This is an excellent, albeit hard-to-answer, question. We tried to address this issue using the proteomic approach and our downstream analyses based on these data. However the exact mechanism likely is multifactorial and might very well depend on many inter-connected signaling pathways. We found that the combination of ATO and Gos reduces many proteins involved in DNA damage response and potently induces DNA damage and followed that line of investigation. Based on our proteomic data we also deduced that this might be in part be mediated by formation of ROS. But to pinpoint this to specific proteins would have been beyond the scope of this manuscript and needs to addressed in due detail in future studies. Nonetheless have now added a more extensive discussion of the effects of ATO and how they might relate to the combination treatment.
However, the data quality and the presentation of the experiments is fine and I feel that the data as presented is still informative even without fulfilling these issues, which would have raised the impact significantly.
Answer: Thank you for the positive assessment of our work.
Submission Date
28 December 2018
Date of this review
19 Feb 2019 14:07:50
References:
Alimoghaddam, K. (2014). "A review of arsenic trioxide and acute promyelocytic leukemia." Int J Hematol Oncol Stem Cell Res 8(3): 44-54.
Beauchamp, E. M., Ringer, L., Bulut, G., Sajwan, K. P., Hall, M. D., Lee, Y. C., Peaceman, D., Ozdemirli, M., Rodriguez, O., Macdonald, T. J., Albanese, C., Toretsky, J. A. and Uren, A. (2011). "Arsenic trioxide inhibits human cancer cell growth and tumor development in mice by blocking Hedgehog/GLI pathway." The Journal of clinical investigation 121(1): 148-160.
Bushunow, P., Reidenberg, M. M., Wasenko, J., Winfield, J., Lorenzo, B., Lemke, S., Himpler, B., Corona, R. and Coyle, T. (1999). "Gossypol treatment of recurrent adult malignant gliomas." Journal of neuro-oncology 43(1): 79-86.
Coutinho, E. M. (2002). "Gossypol: a contraceptive for men." Contraception 65(4): 259-263.
Kanzawa, T., Zhang, L., Xiao, L., Germano, I. M., Kondo, Y. and Kondo, S. (2005). "Arsenic trioxide induces autophagic cell death in malignant glioma cells by upregulation of mitochondrial cell death protein BNIP3." Oncogene 24(6): 980-991.
Kim, J., Aftab, B. T., Tang, J. Y., Kim, D., Lee, A. H., Rezaee, M., Chen, B., King, E. M., Borodovsky, A., Riggins, G. J., Epstein, E. H., Jr., Beachy, P. A. and Rudin, C. M. (2013). "Itraconazole and arsenic trioxide inhibit Hedgehog pathway activation and tumor growth associated with acquired resistance to smoothened antagonists." Cancer cell 23(1): 23-34.
Lan, L., Appelman, C., Smith, A. R., Yu, J., Larsen, S., Marquez, R. T., Liu, H., Wu, X., Gao, P., Roy, A., Anbanandam, A., Gowthaman, R., Karanicolas, J., De Guzman, R. N., Rogers, S., Aube, J., Ji, M., Cohen, R. S., Neufeld, K. L. and Xu, L. (2015). "Natural product (-)-gossypol inhibits colon cancer cell growth by targeting RNA-binding protein Musashi-1." Mol Oncol 9(7): 1406-1420.
Voss, V., Senft, C., Lang, V., Ronellenfitsch, M. W., Steinbach, J. P., Seifert, V. and Kogel, D. (2010). "The pan-Bcl-2 inhibitor (-)-gossypol triggers autophagic cell death in malignant glioma." Molecular cancer research : MCR 8(7): 1002-1016.
Round 2
Reviewer 1 Report
Concerns have been addressed.